# Efficient 3D printing via photooxidation of ketocoumarin based photopolymerization

Xiaoyu Zhao [1,2], Ye Zhao [1,2], Ming-De Li[3], Zhong'an Li[1], Haiyan Peng [1,2 ✉], Tao Xie [4] & Xiaolin Xie[1,2 ✉]

Photopolymerization-based three-dimensional (3D) printing can enable customized manufacturing that is difficult to achieve through other traditional means. Nevertheless, it remains challenging to achieve efficient 3D printing due to the compromise between print speed and resolution. Herein, we report an efficient 3D printing approach based on the photooxidation of ketocoumarin that functions as the photosensitizer during photopolymerization, which can simultaneously deliver high print speed (5.1 cm h$^{-1}$) and high print resolution (23 μm) on a common 3D printer. Mechanistically, the initiating radical and deethylated ketocoumarin are both generated upon visible light exposure, with the former giving rise to rapid photopolymerization and high print speed while the latter ensuring high print resolution by confining the light penetration. By comparison, the printed feature is hard to identify when the ketocoumarin encounters photoreduction due to the increased lateral photopolymerization. The proposed approach here provides a viable solution towards efficient additive manufacturing by controlling the photoreaction of photosensitizers during photopolymerization.

[1] Key Lab for Material Chemistry of Energy Conversion and Storage, Ministry of Education, School of Chemistry and Chemical Engineering, Huazhong University of Science and Technology (HUST), Wuhan, China. [2] National Anti-Counterfeit Engineering Research Center, HUST, Wuhan, China. [3] Key Laboratory for Preparation and Application of Ordered Structural Materials of Guangdong Province, Department of Chemistry, Shantou University (STU), Shantou, China. [4] State Key Laboratory of Chemical Engineering, College of Chemical and Biological Engineering, Zhejiang University (ZJU), Hangzhou, China. ✉email: hypeng@hust.edu.cn; xlxie@hust.edu.cn

Three-dimensional (3D) printing represents an additive manufacturing approach in which 3D objects are created with the aid of computer designs. The most powerful advantage of 3D printing is the capability of producing arbitrarily sophisticated structures that are impossible or very challenging to achieve through other traditional manufacturing methods[1–8]. Furthermore, it can integrate predesigned functions into the 3D-printed objects as needed[9–11]. So far, 3D printing has enabled a broad range of promising engineering applications, including solar evaporator[12], energy storage[13], mechanical dissipation[14], chemical reactors[15], microfluidic devices[16], and biomaterials[17,18]. Nevertheless, the compromise between the print speed and resolution still constrains the print efficiency[1,8,9].

3D printing is most typically implemented in a layer-wise fashion[9], despite the successful demonstration of volumetric additive manufacturing reported recently[6,17]. Among various 3D printing techniques, photopolymerization-based digital light processing (DLP) has attracted particular attention[1–4,9,10,12–16], due to that the photopolymerization kinetics can be precisely controlled spatially and temporally[19–28]. In this manner, 3D printing can be well controlled with rationally designed monomers[7,29–31], photoinitiating systems[32–34], chain transfer agents[35], oxygen inhibition layers[2], mobile liquid interfaces[1], and hardwares[6,17,36]. However, the light transmitted through the solidified polymer layers is highly likely to trigger undesired lateral photopolymerization during the printing of subsequent layers and therefore leads to deteriorated print resolution[1,9]. To confine the light penetration and thus to improve the print resolution, a considerable mass of nonreactive light absorbers (also known as opaquing agents, e.g., Sudan Orange G[3], Sudan I[16], and Tinuvin 171[5]) are always required[9,37]. Nonetheless, such improvement comes at the expense of print speed due to that a large amount of light energy is consumed by the nonreactive light absorbers rather than by the photosensitizers[8]. Light absorption by photosensitizers, which is a prerequisite for the conversion of light energy to chemical energy, is critical for photopolymerization-based 3D printing. Hereto, it remains challenging to develop an efficient 3D printing approach with both high print speed and resolution.

Very recently, Hawker and coworkers have proposed an interesting solution to address the above challenge by using photochromic dyes (e.g., diarylethenes) as reactive light absorbers to selectively confine the light penetration[8]. During 3D printing, diarylethenes in the closed form absorb light and subsequently transform into the open form. The open form is colorless so that the photosensitizers gain the visible light energy to trigger the photopolymerization. Based on this elegant design, as high as 50 cm h$^{-1}$ of print speed and 100 μm of print resolution can be achieved using the one-step printing method. Nonetheless, it remains a great challenge to simultaneously improve the print speed and resolution on common DLP hardware.

Alternatively, it is also rational to resolve the above problem by designing new photoinitiating systems. For instance, by using a reversible addition fragmentation chain transfer (RAFT) agent as the reactive light absorber, Boyer and coworkers[38–41] have achieved a print speed of 9.1 cm h$^{-1}$ while the print resolution is reported to be 200 μm[38]. Allonas and coworkers also reported a resolution of 100 μm but with a low print speed of 1.8 cm h$^{-1}$ using other three-component photoinitiating systems[42,43], wherein the photosensitizer functions as the reactive light absorber. Nonetheless, further increasing the print resolution remains a challenge.

Herein, we demonstrate an efficient 3D printing approach by using ketocoumarin as the photosensitizer (also a reactive light absorber), attributed to (1) ketocoumarins are attractive and distinct photosensitizers with high molar extinction coefficients and high intersystem crossing coefficients[44], which would promote the light energy efficiency, photoinitiating efficiency, and print speed, (2) ketocoumarins exhibit high photochemical stability in a solidified polymer to prevent photobleaching and thus can afford a low light penetration depth[45,46], which would help increasing the print resolution, and (3) ketocoumarins would generate products with high molar extinction coefficients after photoreaction, which help boosting the print resolution as well. To demonstrate a proof of concept, we used one ketocoumarin compound, i.e., 3,3′-carbonylbis(7-diethylaminocoumarin) (KCD), as the photosensitizer for efficient 3D printing. Excitingly, an attractive print speed (5.1 cm h$^{-1}$) and high print resolution (23 μm) were simultaneously achieved on a common bottom-up 3D printer. Furthermore, the light energy efficiency increased 12 times in comparison with traditional systems based on nonreactive light absorbers. Deep insight into the mechanism shows that the photoreaction process of KCD significantly matters for controlling the 3D printing resolution and only the photooxidation process can enable well-defined 3D printing. The initiating radical and deethylated ketocoumarin were simultaneously generated during the photooxidation of KCD. The initiating radical triggered a rapid photopolymerization for facile 3D printing, while the deethylated ketocoumarin ensured high print resolution by confining the light penetration. By comparison, the printed feature could not be identifiable when KCD encountered photoreduction. These findings would open a new door to design efficient additive manufacturing by controlling the photoreaction processes of photosensitizers during photopolymerization.

## Results

**Photooxidation products and mechanism.** Our efficient 3D printing is realized via the photooxidation of ketocoumarin-based photopolymerization. For proof of concept, we employed commercially available KCD as the photosensitizer that was initially reported by Specht and coworkers in 1979[47]. KCD exhibits a high intersystem crossing (ISC) efficiency up to 92%[44] and a large maximum molar extinction coefficient ($\varepsilon_{max}$) of $8.8 \times 10^4$ L mol$^{-1}$ cm$^{-1}$ at 458 nm (see Supplementary Fig. 1), both of which are beneficial for efficient photoreaction of KCD with a coinitiator upon visible light irradiation[44]. As proposed in Fig. 1a, the photooxidation of KCD by 2-(4-methoxyphenyl)-4,6-bis(trichloromethyl)-1,3,5-triazine (TA) was expected to occur in the presence of water (remained in the solvent and monomer, see Supplementary Table 2), thereby producing the initiating radical (i.e., TA·) and deethylated product (i.e., KCD_2). The function of TA· is to initiate a rapid photopolymerization for facile 3D printing[48,49], while KCD_2 can confine the light penetration to improve the print resolution due to its large absorption coefficient in the visible light wavelength region ($\varepsilon_{max} = 4.3 \times 10^4$ L mol$^{-1}$ cm$^{-1}$ at 448 nm, see Supplementary Fig. 2).

To start with, the photooxidation products of KCD by TA were identified. Note that although the ketocoumarin/triazine system and other systems based on ketocoumarin or triazine have been proven to be efficient for mediating the visible light photopolymerization[50–52], the related mechanism and products during the photoreaction have not been fully disclosed, thereby hindering the development of efficient 3D printing. We believe that the photooxidation of KCD by TA is thermodynamically favorable upon visible light irradiation, based on the calculated free energy ($-0.14$ eV) of electron transfer using the Rehm–Weller equation[49]:

$$\Delta G_{ET} = e[E_{ox\_KCD} - E_{red\_TA}] - E_{T\_KCD} \qquad (1)$$

where $E_{ox\_KCD}$ and $E_{red\_TA}$ are the oxidation potential of KCD ($+1.06$ V vs. SCE)[22] and reduction potential of TA ($-1.00$ V

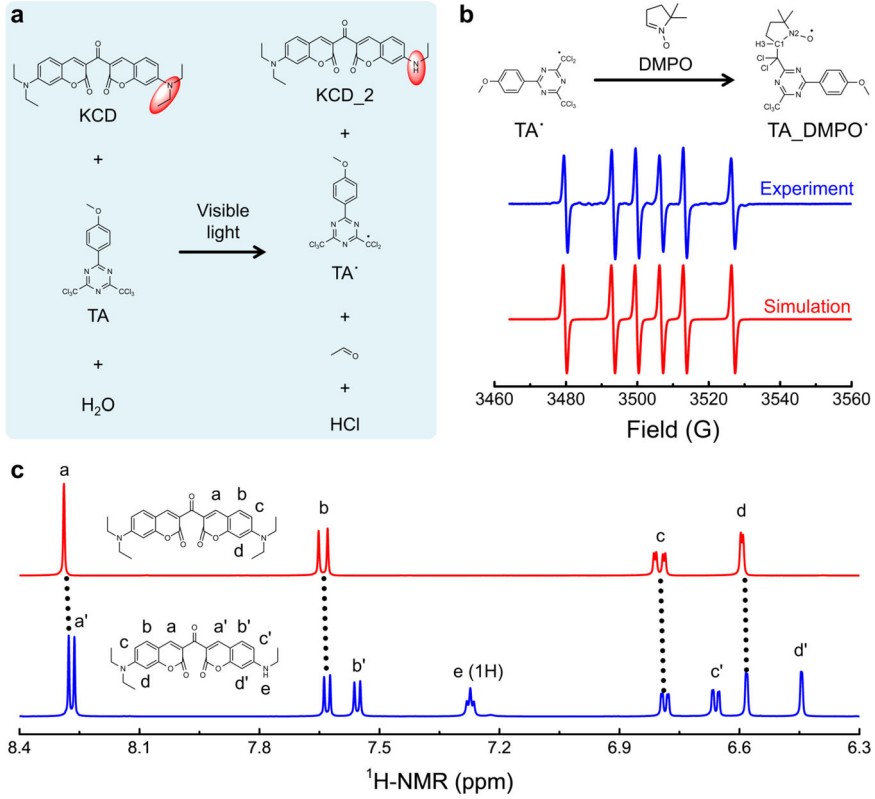

**Fig. 1 Photooxidation products of KCD by TA. a** Proposed products during the photooxidation of KCD by TA upon visible light irradiation in the presence of water (remained in the monomer and solvent). **b** Experimental (blue line) and simulated (red line) electron paramagnetic resonance (EPR, EMXmicro, Bruker) signals that were generated during the photooxidation of KCD by TA. **c** Representative changes of chemical shifts in the $^1$H-NMR spectra (Ascend, Bruker) when converting KCD (red line) to KCD_2 (blue line).

vs. SCE)[49], respectively, and $E_{T\_KCD}$ is the triplet-state energy of KCD (2.20 eV)[44]. Mechanistically, we suspected that KCD encountered twice oxidation when reacting with TA upon visible light irradiation (see Supplementary Fig. 3), which was supported by the transient absorption spectra and density functional theory calculations (see Supplementary Figs. 4 and 5). Wherein, TA was first reduced by accepting one electron from excited KCD. Subsequently, heterolytic cleavage of the C–Cl bond occurred to generate the TA· radical[53], which was evidenced by electron paramagnetic resonance spectroscopy in the presence of 5,5-dimethyl-1-pyrroline N-oxide (DMPO) as the radical stabilizer (Fig. 1b)[22]. No other radicals were detected. The detected radical shows a g-value of 2.0062 along with two hyperfine coupling constants of 20.1 G (H3) and 13.5 G (N2), highly consistent with the previous report[22]. Moreover, the produced KCD_2 was isolated and identified by $^1$H- and $^{13}$C-nuclear magnetic resonance (NMR) spectroscopy (Fig. 1c, and Supplementary Figs. 6 and 7), as well as high-performance liquid chromatography–high-resolution mass spectrometry (HPLC–HRMS, see Supplementary Figs. 8–18). In addition, the generated acetaldehyde ($CH_3CHO$) was also identified using gas chromatography–mass spectrometry (GC–MS, see Supplementary Figs. 19 and 20).

**Print fidelity.** As a control, 3D printing based on the photo-reduction of KCD was also executed. When KCD is photoreduced by amines, e.g., N-phenylglycine (NPG, Fig. 2a) or methyl-diethanolamine (MDEA), the generated amino-alkyl radical and ketyl radical result in a large blueshift of the absorption (~61 nm) compared with KCD[21,22], and thus give rise to a significant decrease in the print resolution.

3D printing experiments show that our photopolymerization-based 3D printing via photooxidation of KCD could enable high print fidelity upon visible light irradiation. To show a proof of concept, we conducted 3D printing with a common benchtop DLP 3D printer (Fig. 2b). The printing was implemented through the bottom-up process, in which the digital light irradiated the printing resin from the bottom. After printing each layer, the platform moved upward to separate the solidified (cross-linked) polymer parts from the surrounding liquid resin and printing vat. Subsequently, the platform was lowered again into the resin to print the next layer. Since both TA and NPG show negligible absorptions above the wavelength of 420 nm (see Supplementary Fig. 21), we added a band-pass filter above the digital light projector to confine the light wavelength (420–780 nm) during printing. Under such a circumstance, only KCD can be directly photoexcited. A monomer mixture composed of N,N-dimethylacrylamide (DMAA), trimethylolpropane ethoxylate triacrylate (TMPEOTA), and pentaerythritol tetraacrylate (PETTA) was used during 3D printing. DMAA can help to dissolve KCD, TA, and NPG. TMPEOTA and PETTA were used to provide cross-linked polymer parts with sufficient strength prior to postcure[9]. However, adhesion of the printed polymer parts to the printing vat should be avoided[1,2]. In the typical 3D printing method, the weight ratio of DMAA, TMPEOTA, and PETTA was optimized to be 3.5:5.0:1.5, which however can be changed depending on the KCD concentration (see Supplementary Fig. 22). Moreover, note that monomers such as 1,6-hexanediol diacrylate (HDDA), poly (ethylene glycol) diacrylate (PEGDA, $M_n$ ~200), and bisphenol A glycerolate diacrylate (Epoxy acrylate) can also be used for successful 3D printing (see Supplementary Fig. 23). Postcure was conducted to consume the unreacted monomers in the polymer

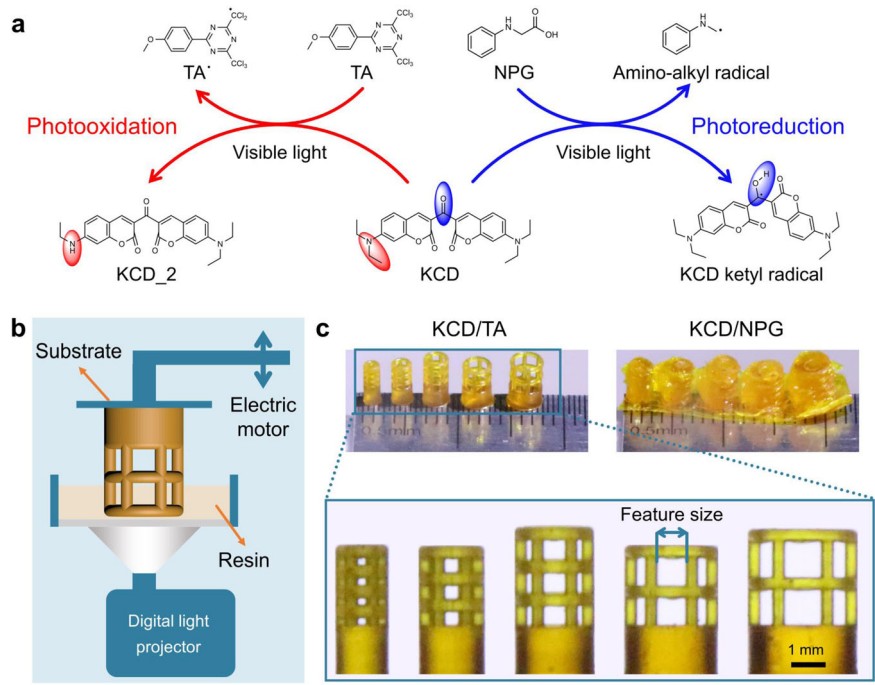

**Fig. 2 Comparison of print fidelity between KCD/TA and KCD/NPG systems. a** Schematic illustration of the photooxidation of KCD by TA and photoreduction by NPG upon visible light irradiation, respectively. **b** Schematic illustration of the bottom-up DLP 3D printing. **c** 3D-printed objects photomediated by KCD/TA and KCD/NPG, respectively, which were printed using the bottom-up DLP 3D printer (Titan 2, Kudo3D). The 3D-printed objects were cleaned with ethanol and then postcured under UV flood for 10 min. Color of the printed objects comes from KCD and KCD_2.

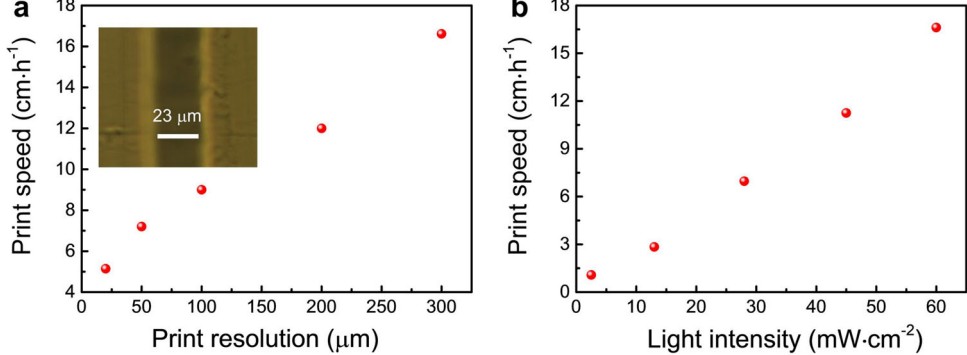

**Fig. 3 High print resolution and speed enabled by photooxidation of KCD.** Print speed against **a** print resolution and **b** light intensity of 420–780 nm light. Inset of **a** Optical microscope image (Axio Scope A1, Carl Zeiss) of the 3D-printed line photomediated by the KCD/TA system.

parts as suggested[9,40]. The double-bond conversion of the printing resin was increased from 36% to 95% upon postcure (see Supplementary Fig. 24). The tensile strength and tensile modulus of printed objects were increased from $21 \pm 2$ MPa and $112 \pm 10$ MPa to $40 \pm 3$ MPa and $260 \pm 15$ MPa, respectively, while the elongation at break was decreased from $36 \pm 3\%$ to $29 \pm 2\%$ (see Supplementary Fig. 25, Supplementary Table 3).

We initially evaluated the print fidelity by printing an array of open cylinders with square lattices. The diameter of each grid line of the square lattice was set as 400 μm in the computer-aided design model, while the feature size of the square lattice was varied from 200 to 1000 μm in 200 μm increments. Excitingly, the printed open cylinders agree well with those predesigned when employing our 3D printing based on the photooxidation of KCD (Fig. 2c and Supplementary Fig. 26). By comparison, the print fidelity showed a dramatic decrease when KCD encountered photoreduction in the presence of amines (e.g., NPG and MDEA). In addition, no printed feature structure can be identified if 3D printing was photomediated by commercially

available photoinitiating systems such as Irgacure 784/benzoyl peroxide and camphorquinone/NPG (see Supplementary Fig. 27), attributed to the increased lateral photopolymerization (see Supplementary Note 1). It is well known that the undesired lateral photopolymerization results in microparticulate solids in the printing resin, which then tend to connect all the printed parts and thus reduce the print fidelity[1].

**Print resolution and speed**. Both high print resolution and print speed have been successfully enabled in our photopolymerization-based 3D printing that is mediated by the photooxidation of KCD. In the absence of KCD or light irradiation, 3D printing was very hard to proceed due to the negligible monomer conversion under those conditions (see Supplementary Fig. 28). We then evaluated the print resolution by printing a line with a width of 1 pixel (23 μm)[54], showing a high print resolution (23 μm) achieved upon exposure to 60 mW cm$^{-2}$ of 420–780 nm light irradiation (Fig. 3a).

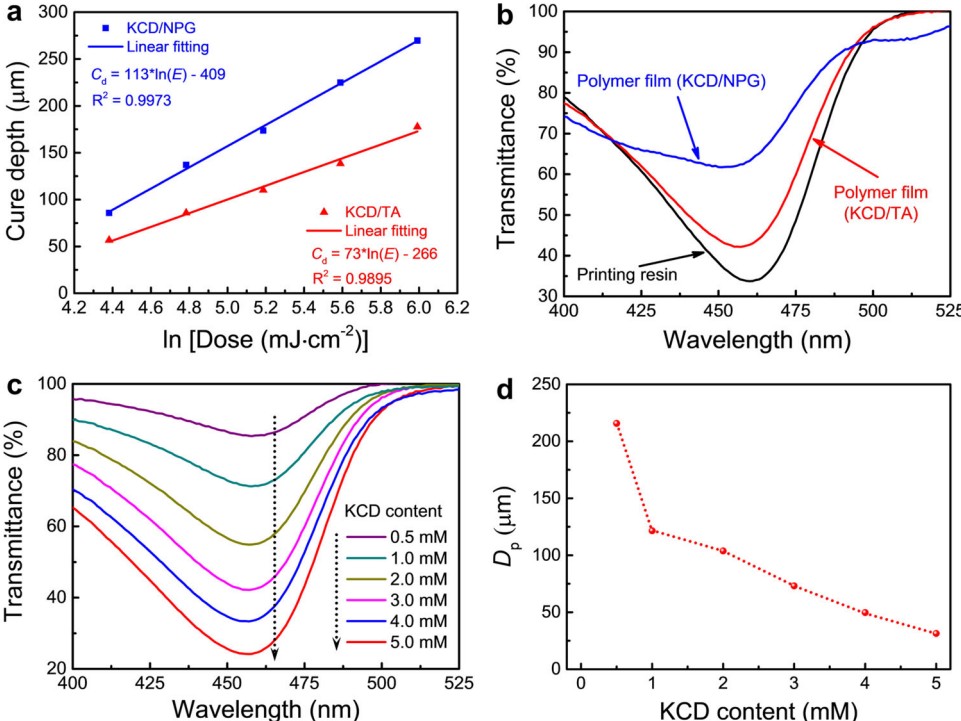

**Fig. 4 Performance of the printing resin. a** Cure depth ($C_d$) of the printing resin against irradiation dose ($E$) at 460 nm. **b** Light transmittance measured by an ultraviolet–visible spectrometer (Revolution 220, Thermo Fisher Scientific) of the polymer films and printing resin with a thickness of 20 μm. **c** Light transmittance of the polymer films with a thickness of 20 μm against KCD content. **d** $D_p$ against KCD content.

Meanwhile, an attractive print speed of 5.1 cm h$^{-1}$ can be achieved by printing each 20 μm thick layer in 1.4 s. It is worth noting that the print speed does not count the moving time of the platform[40,41]. In addition, the print speed was increased from 5.1 to 16.6 cm h$^{-1}$ when the print resolution was varied from 23 to 300 μm (Fig. 3a), and it was decreased from 16.6 to 1.1 cm h$^{-1}$ when the light intensity was decreased from 60 to 2.5 mW cm$^{-2}$ (Fig. 3b).

Elegant designs of hardware and printing resins have been concerned to promote the print speed and print resolution. With respect to the hardware, for instance, DeSimone et al.[2] and Mirkin et al.[1] developed continuous 3D printing setups to increase the print speed to 2.5 cm h$^{-1}$ (resolution: 50 μm) and 43.2 cm h$^{-1}$ (resolution: 300 μm), respectively. Zheng and coworkers used the liquid-crystal on-silicon chip (LCoS) and lens to reduce the pixel size of the digital light engine and thus increased the print resolution to 10 μm[55,56]. Lee and coworkers used the lens to increase the print speed and resolution to 32.7 cm h$^{-1}$ and 100 μm, respectively[57]. Distinctly, in the aspect of printing resins, the highest print resolution (18 μm) was reported by Nordin and coworkers with a print speed of 2.4 cm h$^{-1}$ using a common DLP 3D printer[58]. Furthermore, the 3D printing systems based on RAFT agents[38–41] and three-component photoinitiating systems[42,43] produced print speeds/resolutions of 9.1 cm h$^{-1}$/200 μm and 1.8 cm h$^{-1}$/100 μm on a common DLP 3D printer. In contrast, our KCD/TA system can produce a 12-times higher light energy efficiency ($\varphi = 24\%$) than that of the commonly used nonreactive light-absorber-based system ($\varphi = 2\%$, see Supplementary Notes 2 and 3), leading to an attractive print speed of 5.1 cm h$^{-1}$ and a high print resolution of 23 μm. Note that the print speed is also 2.1 times faster than that of the above mentioned state-of-the-art 3D printing approaches using the common DLP hardware while maintaining comparable print resolution (see Supplementary Table 4).

**Performance of the printing resin**. High print resolution results from the decreased light penetration depth ($D_p$) during 3D printing, which can be calculated from the following equation[9]:

$$C_d = D_p \times \ln(E) - D_p \times \ln(E_c) \qquad (2)$$

where $C_d$ and $E$ denote the cure depth and irradiation dose during 3D printing, respectively, and $E_c$ is the critical irradiation dose of the printing resin. As illustrated in Fig. 4a, despite the identical KCD concentration (3 mM), the calculated $D_p$ (73 μm) of the KCD/TA system is 1.5 times smaller than that of the KCD/NPG system, promising much better print resolution. We believe that this diffference is dominated by the light transmittance of the solidified polymer layers. To provide a quantitative proof, we characterized the light transmittance of polymer films in a thickness of 20 μm. The thickness was identical to that of each printed layer during 3D printing. The results show that the light transmittance of all solidified polymer films is increased compared to the liquid printing resin because of the photoreaction of KCD (Fig. 4b). Within the visible light wavelength region, the minimum light transmittance is 62% and 42% for the polymer films (thickness: 20 μm) photomediated by KCD/NPG and KCD/TA, respectively, indicating a decreased light penetration for the latter. In addition, because of the decreased light transmittance of the polymer film with an increase in KCD concentration (Fig. 4c), the $D_p$ was found to be decreased significantly (Fig. 4d, Supplementary Fig. 29).

Although visible light-sensitive printing resins are of high reactivity, our printing resin can be stored in a brown-color vial under dark or ambient conditions for at least 7 days without detectable changes in the appearance, ground-state absorption, or viscosity (see Supplementary Figs. 30 and 31). However, when the printing resin was stored in a transparent vial under ambient conditions, photopolymerization will occur and induce a dramatic increase in the viscosity.

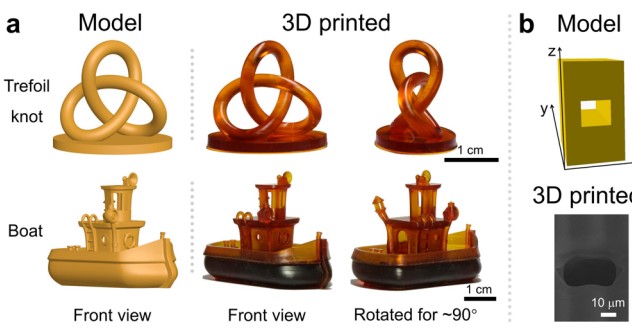

**Fig. 5 Models and corresponding 3D-printed objects. a** Pictures to show the 3D-printed trefoil knot and boat based on the KCD/TA system. The 3D-printed objects were rinsed with ethanol and postcured under UV flood (50 mW cm⁻²) for 10 min. Color of the printed objects comes from KCD and KCD_2. **b** Model of the predesigned microscale hole and the corresponding SEM image based on the KCD/TA system.

**Printed 3D objects**. We can further print 3D-sophisticated objects based on our efficient 3D printing through the KCD/TA system-mediated photopolymerization. Fig. 5a shows the successful printing of a trefoil knot (designed according to the reported method[59]) and a boat (model authorized by van-dragon_de from the MakerBot Thingiverse community). The print fidelity is very high in both cases, and the throughputs of trefoil knot and boat were 1.8 and 10.5 cm³ h⁻¹, respectively. To further confirm the high print fidelity for our method, we designed a tiny hole with 20 μm in height and 23 μm (1 pixel) in width, showing that high-fidelity printing can be achieved with the precisely controlled height, despite increasing the width to 35 μm (1.5 pixels) due to the scattering of light (Fig. 5b)[58].

We further demonstrated that KCD can also be photooxidized by onium salts (e.g., iodonium[60] and sulfonium[51]), and thus the proposed paradigm here opens the door to design efficient 3D printing systems by controlling the photoreaction of photosensitizers during photopolymerization.

## Discussion

In summary, we proposed and demonstrated an efficient 3D printing approach via the photooxidation of ketocoumarin (e.g., KCD) based photopolymerization, which enabled both high print speed (5.1 cm h⁻¹) and high print resolution (23 μm). The print speed was much faster than the state-of-the-art 3D printing approaches using the common bottom-up DLP 3D printer without compromising the print resolution. The light energy efficiency was also 12 times higher than traditional systems containing nonreactive light absorbers. Mechanistically, the produced free radicals led to fast photopolymerization for facile 3D printing, while the deethylated KCD confined the light penetration for improving the print resolution. By comparison, the print resolution was dramatically decreased when KCD encountered photoreduction due to that the increased light transmittance of the printed polymer parts led to increased lateral photopolymerization. The proposed 3D printing approach here based on the photooxidation of ketocoumarin paves the way for efficient additive manufacturing by controlling the photoreaction of photosensitizers during photopolymerization.

## Methods

**Materials**. N,N-Dimethylacrylamide (DMAA, purity: 98%), 5,5-dimethyl-1-pyrroline N-oxide (DMPO, purity: 97%), 2-(4-methoxyphenyl)-4,6-bis(trichloromethyl)-1,3,5-triazine (TA, purity: 98%), pentaerythritol tetraacrylate (PETTA, purity: >80%), and 1,6-hexanediol diacrylate (HDDA, purity: >85%) were purchased from TCI Chemicals. Toluene (spectrometric grade, purity: 99.7%), N-phenylglycine (NPG, purity: 97%), acetonitrile (purity: AR), trimethylolpropane ethoxylate triacrylate (TMPEOTA, average $M_n$ ~692), dichloromethane (DCM,

purity: AR), ethylacetate (EA, purity: AR), ammonia water (liquid chromatography/mass spectrometry (LC/MS) grade), and poly(ethylene glycol) diacrylate (PEGDA, $M_n$ ~200) were received from Aladdin, China. Bisphenol A glycerolate diacrylate (Epoxy acrylate) was received from Sigma-Aldrich. Camphorquinone (CQ, purity: 99%) was received from Alfa Aesar. Irgacure 784 (purity: 99%) was obtained from Alfa Chem, China. Benzoyl peroxide (BPO) and methyldiethanolamine (MDEA, purity: 99%) were received from Innochem Co., Ltd., China. 3,3′-Carbonylbis(7-diethylaminocoumarin) (KCD, purity: 99%) was obtained from Acros Organics. DMSO-$d_6$ (purity: 99.8 atom%D) was purchased from J&K Scientific, China. Acetonitrile (AcCN) and water in LC/MS grade for high-performance liquid chromatography–high-resolution mass spectrometry (HPLC–HRMS) characterization were purchased from Thermo Fisher Scientific. NPG was recrystallized from hot water in the presence of activated charcoal before use, while other chemicals were used directly without further purification.

**Gas chromatography-mass spectrometry (GC-MS)**. Acetaldehyde (CH₃CHO) was expected to generate during the photooxidation of KCD by TA, which was characterized using GC–MS. First, KCD (1 mM) and TA (1 mM) were homogeneously dissolved in toluene upon bulk-ultrasonication at 298 K for 30 min. Then the homogenous solution was purged with argon gas for 30 min to remove the oxygen, followed by white light irradiation (wavelength: 420–780 nm) for 30 min with an intensity of 20.0 mW cm⁻². After removing the solid by filtration, the residue was characterized using a GC/MS setup (5977B MSD, Agilent Technology, United States) with the Ultra Inert column (HP-5MS). The applied parameters were set as follows: carrier gas, high-purity helium; injection port temperature, 573 K; distribution ratio of flow, 10/1; injection sample volume, 1 μL; carrier gas flow rate, 1 mL min⁻¹; ion source, electron ionization.

**High-performance liquid chromatography–high-resolution mass spectrometry (HPLC–HRMS)**. The main products generated during the photooxidation of KCD (1 mM) by TA (1 mM) in toluene were identified using HPLC–HRMS on an LC/MS setup (Orbitrap, Thermo Fisher Scientific, United States) with a C18 column. Prior to characterization, toluene was removed under vacuum and the residual was dissolved in AcCN. The mobile phase was the mixture of AcCN and water in LC/MS grade. The pH of the mobile phase was tuned to be 9.0 with ammonia water in order to improve the chromatography efficiency. The flow rate was 0.2 mL min⁻¹. The volume fraction of AcCN in the mobile phase was optimized to be 30% from 0 to 12 min, which was then increased to 95% from 12 to 20 min. The ion source was atmospheric pressure chemical ionization. Chemical species in the mobile phase were probed by light with wavelengths of 460 and 226 nm, respectively, at the temperature of 303 K.

**Nuclear magnetic resonance (NMR) spectroscopy**. KCD_2 as one of the primary photooxidation products was also identified by NMR after isolation through silica column chromatography (eluent: DCM/EA with a volume ratio of 50/1). Both ¹H- and ¹³C-NMR characterizations were conducted on a 600 MHz NMR spectrometer (Ascend, Bruker, Germany) at 298 K. DMSO-$d_6$ was used as the solvent.

**Ground-state absorption spectroscopy**. Ground-state absorptions were characterized on one ultraviolet–visible (UV–vis) spectrometer (Revolution 220, Thermo Fisher Scientific, United States). AcCN was used as the solvent. The concentrations of KCD, KCD_2, TA and NPG were controlled to be 10 μM.

**Characterization of water content**. The water contents in toluene, DMAA, TMPEOTA, and PETTA were measured using a Karl Fischer moisture titrator (870 KF Titrino plus, Metrohm, Switzerland).

**Electron paramagnetic resonance (EPR) spectroscopy**. EPR spectroscopy was conducted on a spectrometer (EMXmicro, Bruker, Germany) to identify the radical species during photoreaction. A homogeneous solution was prepared by dissolving KCD (175 μM), TA (3.5 mM), and DMPO (the radical capturing agent, 3.5 mM) in toluene upon bulk ultrasonication (298 K, 30 min). The concentrations of KCD and TA were maintained to be identical to that during the nanosecond transient absorption characterization, respectively. Prior to measurement, oxygen was removed by continuous argon gas purge. Then, the solution was loaded in an EPR columnar quartz cell that was subsequently placed into the spectrometer. Radicals were generated upon exposure to light irradiation under a high-pressure mercury lamp and their signals were in situ recorded. The light wavelength was confined to be 420–780 nm. The applied parameters were set as follows: center field, 3514.2 Gauss (G); sweep width, 100 G; microwave power, 2.0 mW; modulation frequency, 100 kHz; modulation amplitude, 1 G; receiver gain, 30 dB; conversion time, 15 ms; time constant, 0.01 ms. The computer program EasySpin[61] was employed to fit the recorded signals.

**Photopolymerization kinetics**. Photopolymerization kinetics upon visible light irradiation (wavelength: 460 nm) was analyzed by real-time Fourier transform-infrared spectroscopy (RT-FTIR, Vertex 80, Bruker, Germany). Plastic spacers were placed between two NaCl salt plates to control the sample thickness (20 μm, which

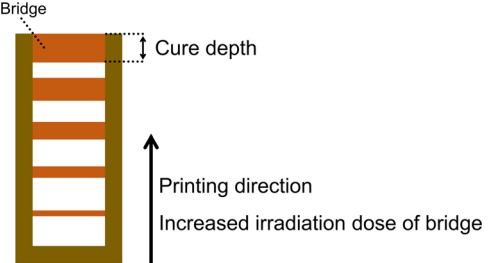

**Fig. 6 Schematic illustration of the cure-depth characterization.** A bridge model was printed for characterization by increasing the irradiation dose along the printing direction.

was the same as the layer thickness during 3D printing). The light intensity was 1.2 mW cm$^{-2}$ (identical to the light intensity when characterizing the penetration depth and critical irradiation dose) after considering the light transmittance of the NaCl plate. The absorption band associated with the C–H rocking vibration of vinyl groups[22] was monitored to characterize the double-bond conversion ($\alpha$) of acrylates[62]:

$$\alpha = 1 - \frac{A_t}{A_0} \tag{3}$$

where $A_t$ and $A_0$ represented the absorption areas at the irradiation time $t$ and before light irradiation, respectively.

**Light transmittance of polymer films.** Light transmittance of the photomediated polymer films was measured on a UV–vis spectrometer (Revolution 220, Thermo Fisher Scientific, United States). Before characterization, the printing resin was injected into a glass cell with a gap of 20 μm (same as the layer thickness during 3D printing), followed by light irradiation from the 3D printer (Titan 2, Kudo3D, United States) with a band-pass filter to confine the light wavelength (420–780 nm). The light intensity was 60 mW cm$^{-2}$, and the irradiation duration was 2 s, which were identical to the intensity and duaration used to print each layer during 3D printing. To obtain the exact light transmittance of the polymer film, the light absorbance of each glass slide was measured and subtracted as the background. The concentrations of KCD were 0.5, 1.0, 2.0, 3.0, 4.0, and 5.0 mM, respectively, while the concentrations of NPG and TA were fixed at 50.0 mM, respectively.

**Characterization of the penetration depth.** To characterize the penetration depth ($D_p$), the cure depth of the printing resin was measured against the irradiation dose. The bridge model was used (Fig. 6)[63], and the cure depth was measured using an optical microscope (Axio Scope A1, Carl Zeiss).

**3D printing.** 3D printing was implemented on a cost-effective benchtop DLP printer (Titan 2, Kudo3D, United States). A band-pass filter was employed to confine the incident light so that the output light was 420–780 nm, and the intensity was 60 mW cm$^{-2}$. The 3D-printed objects were washed with ethanol and postcured under UV light (50 mW cm$^{-2}$) for 10 min.

**Field-effect scanning electron microscopy.** To determine the resolution of the 3D-printed microscale objects, samples were characterized by a field-effect scanning electron microscope (SU8010, Hitachi, Japan). Before characterization, samples were adhered to metal plates using conductive tapes, and then a thin layer of gold was sputtered on the surface to improve the image quality during measurement.

## Data availability
All the relevant data supporting the findings of this study are available from the corresponding author upon request.

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

## Acknowledgements

We thank the financial support from the NSFC (51773073 and 52073108), the HUST peak boarding program, and the Fundamental Research Funds for the Central Universities (2019kfyRCPY089). We also thank the technical assistance from the HUST Analytical & Testing Center and Miss Min Lei at the Core Facilities of Life & Sciences (HUST).

## Author contributions

X.Z. conducted the experiments and density functional theory calculations, analyzed the results, and drafted the paper. Y.Z. helped to analyze the results. M.-D.L. performed the transient absorption experiments. Z.A.L. and T.X. participated in revising the paper. H.P. and X.X. supervised the project and edited the paper.

## Competing interests

The authors declare no competing interests.
