## [Peer Review File · Nature Communications]

Reviewer Comments:

Reviewer #1:

The manuscript described DLP based 3D printing via photooxidation of ketocoumarin that can serve as photosensitizer. The main question here is, why ketocoumarin? What is the advantage of using KCD compared to other recently developed photocatalyst? Have the authors tried to use other photosensitizer or photocatalyst?

Authors must provide more solid evidences that this photosensitizer is preferable over other systems (not only NPG) – for example, other photocatalysts/ photosensitizer must be used under similar conditions and the features (speed and resolution) of the printed objects must be compared.

Other comments:

1-The mechanism behind the radical generation should be elaborated in more details

2-Is this radical capable of initiating other change growth polymerization system?

3-Authors must conduct few control experiments and report the speed and resolutions of the printed objects (e.g. without presence of KCD, without light irradiation).

4-The comparison study between KCD/TA and KCD/NPG is so vague. Elaborate more on the “increased lateral photopolymerization” resulted from the use of NPG.

5-Field-effect scanning electron microscopy technique has been used to determine the resolution of 3D printed objects, however, the detail procedure of this technique has not been provided.

6-There are several recent works on 3D printing with unique features and/or high resolution, which need to be explained in the “introduction” section: Polymer Chemistry 11 (3), 641-647; ACS Applied Polymer Materials 2 (2), 782-790; Angewandte Chemie 131 (50), 18122-18131; Nature communications 10.1 (2019): 1-10, Advanced Materials 30.31 (2018): 1800364 and “Rapid High Resolution Visible Light 3D Printing” by Zachariah A. Page

Reviewer #2 (Remarks to the Author):

Zhao et al. presented a fabrication of 3D objects utilizing ketocoumarin photooxidation triggered photopolymerization to achieve high print speed and high print resolution in common 3D printing approach. They also applied this method to fabricate some complex monolith 3D objects. Initiation is always take an important role in photopolymerization to achieve an efficient curing of photosensitive materials and maintain a pre-programmed structure. And what the authors have done is to meet a new photoinitiation procedure and promise a faster and more accurate polymerization. I am convinced this paper will have a deep impact to the community. I would also expect further research could apply this sort of initiation procedure to varies 3D printing methods to accelerate the efficient of photopolymerization used in 3D printing. The authors also concluded well prospective developments for the future. There are only a few small changes necessary.

- >1. The manuscript could tell the printing speed under different resolution as well as light intensity.
- >2. Authors should also better comment on the processing time to fabricate objects in fig 6. Maybe authors could show the throughput (cm³/h) of this method.
- >3. Authors could measure the double-bond conversion of printed structures, and it would be better to tell the difference between before postcured and after.
- >4. Authors could try other materials to illustrate the flexibility of this photooxidation process in different photosensitive resins, such as epoxy acrylate or polyurethane diacrylate.
- >5. Some mechanical measurements on the printed structures could be presented to exhibit the robustness of the printed objects.
- >6. Since visible-light sensitive photopolymerization could be of high reactivity, there is a need to evaluate the stability of this photopolymerization system.

Reviewer #3 (Remarks to the Author):

In the manuscript, the authors report an efficient 3D printing approach based on the photooxidation of ketocoumarin that functions as the photosensitizer during photopolymerization so that it can simultaneously deliver high print speed and high print resolution on a common 3D printer. However, in the reviewer's view this work lacks generality, and compared with many previously reported works, the printing speed and resolution are not as high as the authors claim. Therefore, this work is not suitable to a high impact journal like Nature Communications. Following are the reasons.

1. All the printing characterizations were conducted using a specific polymer resin composed of DMAA, TEPEOTA, PETTA. Is the KCD applicable to other widely used polymer resins such as PEGDA, HDDA, tBA, Acrylamide, GelMA, etc.?
2. In Figure 3c, the authors claim that their 3D printing enabled by photooxidation of KCD has highest printing speed while keeping the highest resolution. However, apparently, the author ignored many great papers that reported high resolution and high printing approaches. For example, Walker et al. reported "Rapid, large-volume, thermally controlled 3D printing using a mobile liquid interface" on Science 366 360–4 which is able to print 3D structures with the printing speed of 120 $\mu\text{m/s}$ (43.2 cm/h) at the printing resolution of 100 μm ; Tumbleston et al. reported "Continuous liquid interface production of 3D objects" on Science (2015) 347 1349-1351 which is able to print 3D structures with the printing speed of 10 cm/h at the printing resolution of 50 μm . Moreover, there

are many papers that report much higher printing resolution such as Zheng et al. Design and optimization of a light-emitting diode projection micro-stereolithography three-dimensional manufacturing system *Rev. Sci. Instrum.* (2012) 83 125001; Zheng et al Ultralight, ultrastiff mechanical metamaterials *Science* (2014) 344 1373–7; Han et al. 4D printing of a bioinspired microneedle array with backward-facing barbs for enhanced tissue adhesion *Adv. Funct. Mater.* (2020) 30 1909197. If the data from those papers are added to Figure 3c, the results of this work will not be impressive in both printing speed and resolution.

3. The approach presented in Figure 3a is not the best one to investigate the printing resolution. To investigate the lateral resolution, researchers usually print a line or a grid structure to measure the linewidth (Kuang et al., Grayscale digital light processing 3D printing for highly functionally graded materials, *Science Advances* (2019), Vol. 5, no. 5, eaav5790). To characterize curing depth and vertical resolution is more challenging, the bridge characterization approach proposed by Han et al. (Rapid multi-material 3D printing with projection micro-stereolithography using dynamic fluidic control, *Additive Manufacturing* (2019) 27 606-615) is a better way to investigate the curing depth and vertical resolution.

4. Moreover, a good control on curing depth is critical for fabricating structure with holes and channels. For example, Nordin et al. report a 3D printer combined with self-developed resin that is able to fabricate $18\ \mu\text{m} \times 20\ \mu\text{m}$ microfluidic flow channels. (Custom 3D printer and resin for $18\ \mu\text{m} \times 20\ \mu\text{m}$ microfluidic flow channels, *Lab Chip*, (2017), 17, 2899–2909). Could authors print microfluidic channels with smaller feature than this paper?

5. According to the manuscript, the commercial 3D printer Kudo 3D was used to printed characterize the printing resolution and print 3D structures. However, due to the manual of Kudo 3D, its highest optical printing resolution is $23\ \mu\text{m}$. This work achieves the resolution better than $23\ \mu\text{m}$. How did the authors achieve that?

6. According to Figure 3c, the photosensitizer used in this paper enables 3D printing of structures with high resolution (up to $20\ \mu\text{m}$). Figure 6 should demonstrate some printed structures that matches such high resolution.

Point-by-Point Response

Reviewer 1#:

The manuscript described DLP based 3D printing via photooxidation of ketocoumarin that can serve as photosensitizer. The main question here is, why ketocoumarin? What is the advantage of using KCD compared to other recently developed photocatalyst? Have the authors tried to use other photosensitizer or photocatalyst?

Authors must provide more solid evidences that this photosensitizer is preferable over other systems (not only NPG) – for example, other photocatalysts/ photosensitizer must be used under similar conditions and the features (speed and resolution) of the printed objects must be compared.

Response: We thank the referee for the valuable comment. For the DLP 3D printing, a low light penetration depth is critical to promote the print resolution. In previous works, nonreactive light absorbers have been typically used to decrease the light penetration depth and thus to improve the print resolution. However, the print speed and light energy efficiency can be dramatically decreased for those systems due to that a large amount of light energy are consumed by the nonreactive light absorbers rather than the photosensitizer. Consequently, the light energy efficiency is only ~2%.

In sharp contrast, ketocoumarins (*e.g.*, KCD) are attractive and distinct photosensitizers to overcome the drawbacks of the systems based on nonreactive light absorbers. (1) KCD shows high photochemical stability in polymer films to prevent photobleaching and thus can afford a low light penetration depth. (2) Its high molar extinction coefficient up to $8.8 \times 10^4 \text{ L} \cdot \text{mol}^{-1} \cdot \text{cm}^{-1}$ not only helps to decrease the light

penetration depth, but also helps to promote the photoinitiating efficiency and light energy efficiency during 3D printing. (3) Its high intersystem crossing coefficient up to 92% can provide a high photoinitiating efficiency for increasing the 3D printing speed. (4) The photooxidized product of KCD still holds a high molar extinction coefficient with a small blue shift of the absorption peak, which is beneficial for the high print speed as well as print resolution. Consequently, both high print speed ($5.1 \text{ cm}\cdot\text{h}^{-1}$) and high print resolution ($23 \text{ }\mu\text{m}$) are achieved based on the photooxidation of KCD. More excitingly, the light energy efficiency is 12 times higher than that of traditional systems based on nonreactive light absorbers.

Supplementary Figure 27. Printed feature size that was measured under an optical microscope (Axio Scope A1, Carl Zeiss) against that designed with varied photosensitizers and co-initiators. Averages with standard deviations (SD) are presented by measuring 6 square lattices for each entry.

We also conducted more control experiments as suggested by the referee. As clearly displayed in **Supplementary Figure 27**, no feature size can be identifiable when employing the photoinitiating systems such as Irgacure 784/benzoyl peroxide (BPO) and camphorquinone (CQ)/NPG for printing the square lattices although we

changed the print speed from 0.4 to 3.6 cm·h⁻¹. We also employed the KCD/methyldiethanolamine (MDEA) as the photoinitiating system for 3D printing, but the results indicate that the feature size of 200 μm cannot be identifiable due to that the photoreduction of KCD occurred as did in the KCD/NPG system.

To address the referee's concern, we have added more discussions in the revision.

- A. In the Introduction part, we have given a paragraph to review the recent advance of 3D printing related to the photoinitiating systems.
- B. In the Introduction part, we have added the following sentences: “we demonstrated an efficient 3D printing approach by using ketocoumarin as the photosensitizer (also a reactive light absorber), attributed to (1) ketocoumarins are attractive and distinct photosensitizers with high molar extinction coefficients and high intersystem crossing coefficients,⁴⁴ which would promote the light energy efficiency, photoinitiating efficiency and print speed; (2) ketocoumarins exhibit high photochemical stability in a solidified polymer to prevent photobleaching and thus can afford a low light penetration depth,^{45,46} which would help increasing the print resolution; (3) ketocoumarins would generate products with high molar extinction coefficients after photoreaction, which help boosting the print resolution as well. To demonstrate a proof of concept, we used one ketocoumarin compound, *i.e.*, 3,3'-carbonylbis(7-diethylaminocoumarin) (KCD), as the photosensitizer for efficient 3D printing. Excitingly, an attractive print speed (5.1 cm·h⁻¹) and high print resolution (23 μm) were simultaneously achieved on a common bottom-up 3D printer. Furthermore, the light energy

efficiency increased 12 times in comparison with traditional systems based on nonreactive light absorbers.”

C. In the Results part, we have added the following sentences: “In addition, no printed feature structure can be identified if 3D printing was photomediated by commercially available photoinitiating systems such as Irgacure 784/benzoyl peroxide and camphorquinone/NPG (see **Supplementary Fig. 27**), attributed to the increased lateral photopolymerization (see **Supplementary Note 1**).”

Other comments:

1-The mechanism behind the radical generation should be elaborated in more details

Response: We thank the referee for raising this concern. The detailed mechanism of the photooxidation of KCD by TA and the radical generation process were investigated using transient absorption spectra and density functional theory (DFT) calculations. Corresponding data was given in **Supplementary Figure 3~5**.

To address the referee’s concern, the following sentences were added in the revised main text: “Mechanistically, we suspected that KCD encountered twice oxidation when reacting with TA upon visible light irradiation (see **Supplementary Fig. 3**), which was supported by the transient absorption spectra and density functional theory (DFT) calculations (see **Supplementary Fig. 4, 5**). Wherein, TA was first reduced by accepting one electron from excited KCD. Subsequently, heterolytic cleavage of C-Cl bond occurred to generate the TA[•] radical,⁵³ which was evidenced by electron paramagnetic resonance (EPR) spectroscopy in the presence of

5,5-dimethyl-1-pyrroline *N*-oxide (DMPO) as the radical stabilizer (**Fig. 1b**).^{22,}

2-Is this radical capable of initiating other chain growth polymerization system?

Response: We thank the referee for the question. We demonstrated the chain growth polymerization in the revision using other monomers such as 1,6-hexanediol diacrylate (HDDA), poly(ethylene glycol) diacrylate (PEGDA, $M_n \sim 200$), bisphenol A glycerolate diacrylate (Epoxy acrylate). Wherein, KCD and TA can be well dissolved in the monomers of HDDA and PEGDA, while Epoxy acrylate should be mixed with 30 wt% of DMAA to help dissolving the solids. These monomers can be used for 3D printing that was photomediated by the KCD/TA system. Related discussions have been also added in the revised main text.

Supplementary Figure 23 | Images of 3D printed hollow cylinders using the monomer of HDDA, PEGDA and Epoxy acrylate.

3-Authors must conduct few control experiments and report the speed and resolutions of the printed objects (e.g. without presence of KCD, without light irradiation).

Response: We thank the referee for the valuable suggestion. We conducted the experiments as suggested. Results show that no printing can proceed without KCD or light irradiation, mainly due to the negligible monomer conversion under those conditions.

Supplementary Figure 28 | Monomer conversion mediated by (a) NPG and (b) TA that was measured by RT-FTIR (Vertex 80, Bruker) against irradiation dose of 460 nm light with light intensity of $1.2 \text{ mW}\cdot\text{cm}^{-2}$; (c) Monomer conversion without light irradiation.

To address the referee’s question, the following sentences were added in the revised main text: “In absence of KCD or light irradiation, 3D printing was very hard to proceed due to the negligible monomer conversion under those conditions (see **Supplementary Fig. 28**).”

4-The comparison study between KCD/TA and KCD/NPG is so vague. Elaborate more on the “increased lateral photopolymerization” resulted from the use of NPG.

Response: We thank the referee for raising this concern. Because of the expanded light path of digital light and light scattering by microparticulate solids, lateral photopolymerization usually occurs in the area beyond that predesigned, enlarging the solidified areas and decreasing the print resolution (*Science* 2019, 366, 360). Therefore, it is valuable to decrease the light transmittance of solidified polymer parts and light penetration depth so that the lateral photopolymerization can be depressed and the print resolution can be increased. Within the visible light wavelength region, the minimum light transmittance is 62% and 42% for the polymer films (thickness: 20 μm) photomediated by KCD/NPG and KCD/TA, respectively, indicating a decreased light penetration and weakened lateral photopolymerization for the latter.

Consequently, the feature size of the square lattice is barely identifiable when photomediated by the KCD/NPG system, while the KCD/TA system generates a much higher print resolution. A scheme and one paragraph discussion have been added in the revised Supporting Information for clarity.

Supplementary Figure 32 | Illustration on the lateral photopolymerization of 3D printing.

5-Field-effect scanning electron microscopy technique has been used to determine the resolution of 3D printed objects, however, the detail procedure of this technique has not been provided.

Response: We thank the referee for bringing this issue to our attention. We have added details in the revision: “To determine the resolution of the 3D printed microscale objects, samples were characterized by a field-effect scanning electron microscope (FE-SEM, SU8010, Hitachi, Japan). Before characterization, samples were adhered to metal plates using conductive tapes, and then a thin layer of gold was sputtered on the surface to improve the image quality during measurement.”

6-There are several recent works on 3D printing with unique features and/or high

resolution, which need to be explained in the “introduction” section: Polymer Chemistry 11 (3), 641-647; ACS Applied Polymer Materials 2 (2), 782-790; Angewandte Chemie 131 (50), 18122-18131; Nature communications 10.1 (2019): 1-10, Advanced Materials 30.31 (2018): 1800364 and “Rapid High Resolution Visible Light 3D Printing” by Zachariah A. Page

Response: We thank the referee for the good suggestion and have cited these references in the revised main text.

Reviewer #2:

Zhao et al. presented a fabrication of 3D objects utilizing ketocoumarin photooxidation triggered photopolymerization to achieve high print speed and high print resolution in common 3D printing approach. They also applied this method to fabricate some complex monolith 3D objects.

Initiation is always take an important role in photopolymerization to achieve an efficient curing of photosensitive materials and maintain a pre-programmed structure. And what the authors have done is to meet a new photoinitiation procedure and promise a faster and more accurate polymerization. I am convinced this paper will have a deep impact to the community. I would also expect further research could apply this sort of initiation procedure to varies 3D printing methods to accelerate the efficient of photopolymerization used in 3D printing.

The authors also concluded well prospective developments for the future. There are only a few small changes necessary.

1. The manuscript could tell the printing speed under different resolution as well as light intensity.

Response: We thank the referee for the valuable suggestion. We have added the results in Fig. 3 of the revised main text. A sentence has also been added: “In addition, the print speed was increased from 5.1 cm·h⁻¹ to 16.6 cm·h⁻¹ when the print resolution was varied from 23 μm to 300 μm (Fig. 3a), and it was decreased from 16.6 cm·h⁻¹ to 1.1 cm·h⁻¹ when the light intensity was decreased from 60 mW·cm⁻² to 2.5 mW·cm⁻² (Fig. 3b).”

Fig. 3 High print resolution and speed enabled by photooxidation of KCD. Print speed against (a) print resolution and (b) light intensity of 420-780 nm light. Inset of (a): Optical microscope image (Axio Scope A1, Carl Zeiss) of the 3D printed line photomediated by the KCD/TA system.

2. Authors should also better comment on the processing time to fabricate objects in fig 6. Maybe authors could show the throughput (cm³/h) of this method.

Response: We thank the referee for the good suggestion. The throughput of trefoil knot and boat was calculated to be 1.8 cm³·h⁻¹ and 10.5 cm³·h⁻¹. These values have been added in the revision.

3. Authors could measure the double-bond conversion of printed structures, and it would be better to tell the difference between before postcured and after.

Response: We thank the referee for the good suggestion. We characterized the double-bond conversion (α) of the printing resin after printing and postcuring, which is 36% and 95%, respectively. Related discussions have been added in the revision.

Supplementary Figure 24 | Characteristic peak absorptions of C=C bond before printing, after printing and after postcuring, respectively.

4. Authors could try other materials to illustrate the flexibility of this photooxidation process in different photosensitive resins, such as epoxy acrylate or polyurethane diacrylate.

Response: We thank the referee for the constructive suggestion. Monomers such as 1,6-hexanediol diacrylate (HDDA), poly(ethylene glycol) diacrylate (PEGDA, $M_n \sim 200$), bisphenol A glycerolate diacrylate (Epoxy acrylate) were printed and the results were added in **Supplementary Figure 23** of the revision. Also see the response to Reviewer #1.

5. Some mechanical measurements on the printed structures could be presented to exhibit the robustness of the printed objects.

Response: We thank the referee for the valuable suggestion. We have characterized the stress-strain curves of 3D printed samples before and after postcuring, wherein the tensile strength was increased from 21 ± 2 MPa to 40 ± 3 MPa, the elongation at break was decreased from $36\pm 3\%$ to $29\pm 2\%$, the tensile modulus was increased from 112 ± 10 MPa to 260 ± 15 MPa.

To address the referee's concern, the following sentences were added in the revised main text: "The double-bond conversion of the printing resin was increased from 36% to 95% upon postcure (see **Supplementary Fig. 24**). The tensile strength and tensile modulus of printed objects were increased from 21 ± 2 MPa and 112 ± 10 MPa to 40 ± 3 MPa and 260 ± 15 MPa, respectively, while the elongation at break was decreased from $36\pm 3\%$ to $29\pm 2\%$ (see **Supplementary Fig. 25, Supplementary Table 3**)."

Supplementary Table 3 | Tensile strength, tensile modulus and elongation at break of 3D printed samples before and after postcuring

Entry	Before postcuring			After postcuring		
	Tensile strength (MPa)	Tensile modulus (MPa)	Elongation at break (%)	Tensile strength (MPa)	Tensile modulus (MPa)	Elongation at break (%)
1	19	110	38	44	276	27
2	24	96	34	42	232	27
3	20	124	38	35	265	29
4	22	118	40	39	268	29
5	21	114	32	39	259	34

Average	21±2	112±10	36±3	40±3	260±15	29±2
---------	------	--------	------	------	--------	------

6. Since visible-light sensitive photopolymerization could be of high reactivity, there is a need to evaluate the stability of this photopolymerization system.

Response: We thank the referee for the valuable suggestion. We characterized the stability of this photopolymerization system using ultraviolet-visible spectrophotometry and rheology. When the photopolymerizable system was kept in brown-color vial, its appearance, ground state absorption and viscosity were barely changed after 7 days when stored in dark and ambient conditions. However, photopolymerization will occur when keeping the system in transparent vial and ambient condition, leading to a dramatically increased viscosity.

Supplementary Figure 30 | Appearance of the printing resin when kept in brown-color vial & dark condition, brown-color vial & ambient condition and transparent vial & ambient condition.

Supplementary Figure 31 | (a) Molar extinction coefficient ($\epsilon @ 458 \text{ nm}$) and (b) viscosity of printing resins when kept in brown-color vial & dark condition, brown-color vial & ambient condition and transparent vial & ambient condition.

To address the referee’s concern, the following sentences have been added in the revised main text: “Although visible light sensitive printing resins are of high reactivity, our printing resin can be stored in the brown-color vial under dark or ambient conditions for at least 7 days without detectable changes in the appearance, ground state absorption and viscosity (see **Supplementary Fig. 30, 31**). However, when the printing resin was stored in a transparent vial under ambient conditions, photopolymerization will occur to induce a dramatic increase in the viscosity.”

Reviewer #3:

In the manuscript, the authors report an efficient 3D printing approach based on the photooxidation of ketocoumarin that functions as the photosensitizer during photopolymerization so that it can simultaneously deliver high print speed and high print resolution on a common 3D printer. However, in the reviewer’s view this work lacks generality, and compared with many previously reported works, the printing speed and resolution are not as high as the authors claim. Therefore, this work is not

suitable to a high impact journal like Nature Communications. Following are the reasons.

1. All the printing characterizations were conducted using a specific polymer resin composed of DMAA, TEPEOTA, PETTA. Is the KCD applicable to other widely used polymer resins such as PEGDA, HDDA, tBA, Acrylamide, GelMA, etc.?

Response: We thank the referee for the valuable comment. We have conducted experiments as suggested by using a variety of monomers. Results show that our method is generally applicable. Results are shown in **Supplementary Figure 23** and discussions are added in the main text. Also see the response to Reviewer 1#.

2. In Figure 3c, the authors claim that their 3D printing enabled by photooxidation of KCD has highest printing speed while keeping the highest resolution. However, apparently, the author ignored many great papers that reported high resolution and high printing approaches. For example, Walker et al. reported “Rapid, large-volume, thermally controlled 3D printing using a mobile liquid interface” on Science 366 360–4 which is able to print 3D structures with the printing speed of 120 $\mu\text{m/s}$ (43.2 cm/h) at the printing resolution of 100 μm ; Tumbleston et al. reported “Continuous liquid interface production of 3D objects” on Science (2015) 347 1349-1351 which is able to print 3D structures with the printing speed of 10 cm/h at the printing resolution of 50 μm . Moreover, there are many papers that report much higher printing resolution such as Zheng et al. Design and optimization of a light-emitting diode projection micro-stereolithography three-dimensional manufacturing system Rev. Sci. Instrum. (2012) 83 125001; Zheng et al Ultralight, ultrastiff mechanical metamaterials Science

(2014) 344 1373–7; Han et al. 4D printing of a bioinspired microneedle array with backward-facing barbs for enhanced tissue adhesion *Adv. Funct. Mater.* (2020) 30 1909197. If the data from those papers are added to Figure 3c, the results of this work will not be impressive in both printing speed and resolution.

Response: We thank the referee for bringing the issue to our attention.

As we rechecked, the reference of “Rapid, large-volume, thermally controlled 3D printing using a mobile liquid interface (*Science* 2019, 366, 360)” enabled a print speed of $43.2 \text{ cm}\cdot\text{h}^{-1}$, but the print resolution was $300 \text{ }\mu\text{m}$. The reference of “Continuous liquid interface production of 3D objects (*Science* 2015, 347, 1349)” reported a print speed of $2.5 \text{ cm}\cdot\text{h}^{-1}$ with a resolution of $50 \text{ }\mu\text{m}$. The reference of “Design and optimization of a light-emitting diode projection micro-stereolithography three-dimensional manufacturing system (*Rev. Sci. Instrum.* 2012, 83, 125001)” reported no data of print speed and print resolution about the 3D printed objects, except the resolution of $1.3 \text{ }\mu\text{m}$ for the light source. The reference of “Ultralight, ultrastiff mechanical metamaterials (*Science* 2014, 344, 1373)” reported a minimum printed feature size of $10 \text{ }\mu\text{m}$ without reporting the data of print speed. The reference of “4D printing of a bioinspired microneedle array with backward-facing barbs for enhanced tissue adhesion (*Adv. Funct. Mater.* 2020, 30, 1909197)” reported a print speed of $32.7 \text{ cm}\cdot\text{h}^{-1}$ with a $100 \text{ }\mu\text{m}$ print resolution.

However, the hardware of 3D printers was elaboratively designed to meet the high print speed or high print resolution in these reports. The reference of “*Science* 2015, 347, 1349” and “*Science* 2019, 366, 360” designed a continuous 3D printing

hardware to promote the print speed. The reference of “*Rev. Sci. Instrum.* 2012, 83, 125001” and “*Science* 2014, 344, 1373” used the liquid-crystal-on-silicon chip (LCoS) and lens to reduce the pixel size of the digital light engine and to increase the print resolution, while the reference of “*Adv. Funct. Mater.* 2020, 30, 1909197” used the lens to increase the print resolution. These elegant designs have greatly promoted the print speed or print resolution from the aspect of hardware.

However, considering the broad availability of the hardware, it is worthy to develop other methods to simultaneously increase the print speed and print resolution. Herein, we demonstrated an efficient 3D printing approach from the materials science aspect using a common setup and would provide a new solution to the inherent conflict of print resolution and print speed.

To be honest, the results are hard to compare when using different hardware. However, to address the referee’s concern, we have removed the original Fig. 3 and added a new table in the Supporting Information for comparison. Clearly, our method can provide both high print speed and high print resolution.

Supplementary Table 4 | Reported print speed and print resolution

No.	Print speed (cm·h ⁻¹)	Print resolution (μm)	DLP printer	Reference
1	2.5	50	Continuous 3D printer	[19]
2	43.2	300	Continuous 3D printer	[1]
3	-	10	Equipped with LCoS and lens	[20]
4	32.7	100	Equipped with lens	[21]
5	2.4	18	Common DLP 3D printer	[22]
6	1.8	100	Common DLP 3D printer	[23]
7	1.2	100	Common DLP 3D printer	[24]
8	0.05	-	Common DLP 3D printer	[25]
9	0.13	-	Common DLP 3D printer	[26]
10	1.2	-	Common DLP 3D printer	[27]

11	9.1	200	Common DLP 3D printer	[28]
12	5.1	23	Common DLP 3D printer	This work

3. The approach presented in Figure 3a is not the best one to investigate the printing resolution. To investigate the lateral resolution, researchers usually print a line or a grid structure to measure the linewidth (Kuang et al., Grayscale digital light processing 3D printing for highly functionally graded materials, *Science Advances* (2019), Vol. 5, no. 5, eaav5790). To characterize curing depth and vertical resolution is more challenging, the bridge characterization approach proposed by Han et al. (Rapid multi-material 3D printing with projection micro-stereolithography using dynamic fluidic control, *Additive Manufacturing* (2019) 27 606-615) is a better way to investigate the curing depth and vertical resolution.

Response: We thank the referee for the valuable suggestion. As suggested, a line with a width of 23 μm (1 pixel) was printed to show the print resolution photomediated by the KCD/TA system (Inset of **Fig. 3a** in the revision).

Fig. 3 High print resolution and speed enabled by photooxidation of KCD. Print speed against (a) print resolution and (b) light intensity of 420-780 nm light. Inset of (a): Optical microscope image (Axio Scope A1, Carl Zeiss) of the 3D printed line photomediated by the KCD/TA system.

As suggested by the referee, we also recharacterized the cure depth of the

printing resin against the irradiation dose using the bridge model (**Fig. 6** in the revision). Results show that the penetration depth (D_p) of KCD/NPG system is 113 μm , which is 1.5 times higher than that of the KCD/TA system (**Fig. 4a** in the revision), suggesting a higher print resolution for the latter.

Fig. 6 Illustration on the cure depth characterization using a bridge model.

To address the referee's concern, the following sentences have been added in the revised main text: "We then evaluated the print resolution by printing a line with width of 1 pixel (23 μm),⁵⁴ showing a high print resolution (23 μm) achieved upon exposure to 60 $\text{mW}\cdot\text{cm}^{-2}$ of 420-780 nm light irradiation (**Fig. 3a**)."

The cure depth was also discussed in the revision by using the suggested methods.

4. Moreover, a good control on curing depth is critical for fabricating structure with holes and channels. For example, Nordin et al. report a 3D printer combined with self-developed resin that is able to fabricate 18 $\mu\text{m} \times 20 \mu\text{m}$ microfluidic flow channels. (Custom 3D printer and resin for 18 $\mu\text{m} \times 20 \mu\text{m}$ microfluidic flow channels, Lab Chip, (2017), 17, 2899–2909). Could authors print microfluidic channels with smaller feature than this paper?

Response: We thank the referee for the valuable suggestion. We designed a tiny hole with 20 μm in height and 23 μm (1 pixel) in width. Results show that the hole in high fidelity can be printed using our method despite that the width increases to 35 μm (1.5 pixels) because of the scattering of light (*Lab Chip* 2017, 17, 2899). It is worth noting that the resolution is hard to be further improved due to the limit of the hardware.

To address the referee's concern, we have added the following sentences in the revised main text: "To further confirm the high print fidelity for our method, we designed a tiny hole with 20 μm in height and 23 μm (1 pixel) in width, showing that high fidelity printing can be achieved with the precisely controlled height despite increasing the width to 35 μm (1.5 pixels) due to the scattering of light (**Fig. 5b**).⁵⁸"

Fig. 5 (a) Pictures to show the 3D printed trefoil knot and boat based on the KCD/TA system. The 3D printed objects were rinsed with ethanol and postcured under UV flood ($50 \text{ mW}\cdot\text{cm}^{-2}$) for 10 min. Color of the printed objects comes from KCD and KCD_2. (b) Model of the predesigned microscale hole and the corresponding SEM image based on the KCD/TA system.

5. According to the manuscript, the commercial 3D printer Kudo 3D was used to printed characterize the printing resolution and print 3D structures. However, due to

the manual of Kudo 3D, its highest optical printing resolution is 23 μm . This work achieves the resolution better than 23 μm . How did the authors achieve that?

Response: We thank the referee for bringing this issue to our attention. We recharacterized the print resolution by printing a line with width of 1 pixel as suggested, indicating a print resolution of 23 μm . All related data have been corrected.

6. According to Figure 3c, the photosensitizer used in this paper enables 3D printing of structures with high resolution (up to 20 μm). Figure 6 should demonstrate some printed structures that matches such high resolution.

Response: We thank the referee for the valuable suggestion. We replaced Figure 6 with a new figure including the printed microscale hole, as to be **Fig. 5** in the revised main text.

REVIEWERS' COMMENTS

Reviewer #1 made comments to the editor only and supports publication.

Reviewer #2 (Remarks to the Author):

The authors have corrected paper accordingly. The improved manuscript is now suitable for publication.

Reviewer #3 (Remarks to the Author):

The authors have addressed all the concerns from the reviewer. It is recommended for publication.

Point-by-Point Response

Reviewer #1 made comments to the editor only and supports publication.

Response: The authors gratefully appreciate the positive recommendation from the referee.

Reviewer #2:

The authors have corrected paper accordingly. The improved manuscript is now suitable for publication.

Response: The authors gratefully appreciate the positive recommendation from the referee.

Reviewer #3:

The authors have addressed all the concerns from the reviewer. It is recommended for publication.

Response: The authors gratefully appreciate the positive recommendation from the referee.